# JNK pathway restricts DENV2, ZIKV and CHIKV infection by activating complement and apoptosis in mosquito salivary glands

Avisha Chowdhury[1]☯, Cassandra M. Modahl[1]☯, Siok Thing Tan[1], Benjamin Wong Wei Xiang[2], Dorothée Missé[3], Thomas Vial[2], R. Manjunatha Kini[1]*, Julien Francis Pompon[2,3]*

**1** Department of Biological Sciences, National University of Singapore, Singapore, **2** Emerging Infectious Diseases, Duke-NUS Medical School, Singapore, **3** MIVEGEC, IRD, CNRS, Univ. Montpellier, Montpellier, France

☯ These authors contributed equally to this work.
* dbskinim@nus.edu.sg (RMK); Julien.pompon@ird.fr (JFP)

## Abstract

Arbovirus infection of *Aedes aegypti* salivary glands (SGs) determines transmission. However, there is a dearth of knowledge on SG immunity. Here, we characterized SG immune response to dengue, Zika and chikungunya viruses using high-throughput transcriptomics. We also describe a transcriptomic response associated to apoptosis, blood-feeding and lipid metabolism. The three viruses differentially regulate components of Toll, Immune deficiency (IMD) and c-Jun N- terminal Kinase (JNK) pathways. However, silencing of the Toll and IMD pathway components showed variable effects on SG infection by each virus. In contrast, regulation of the JNK pathway produced consistent responses in both SGs and midgut. Infection by the three viruses increased with depletion of the activator Kayak and decreased with depletion of the negative regulator Puckered. Virus-induced JNK pathway regulates the complement factor, Thioester containing protein-20 (TEP20), and the apoptosis activator, Dronc, in SGs. Individual and co-silencing of these genes demonstrate their antiviral effects and that both may function together. Co-silencing either *TEP20* or *Dronc* with *Puckered* annihilates JNK pathway antiviral effect. Upon infection in SGs, TEP20 induces antimicrobial peptides (AMPs), while Dronc is required for apoptosis independently of TEP20. In conclusion, we revealed the broad antiviral function of JNK pathway in SGs and showed that it is mediated by a TEP20 complement and Dronc-induced apoptosis response. These results expand our understanding of the immune arsenal that blocks arbovirus transmission.

## Author summary

Arboviral diseases caused by dengue (DENV), Zika (ZIKV) and chikungunya (CHIKV) viruses are responsible for large number of death and debilitation around the world. These viruses are transmitted to humans by the mosquito vector, *Aedes aegypti*. During

**Data Availability Statement:** All transcriptomics data are available at NCBI accessions: SRR8921123-8921132.

**Funding:** This work was funded by the Tier-3 grant from the Ministry of Education, Singapore (MOE 2015-T3-1-003) and the Duke-NUS Signature Research Programme funded by the Agency for Science, Technology and Research (A*STAR), Singapore, and the Ministry of Health, Singapore. The funders had no role in study design, data collection and analysis, decision to publish, or preparation of the manuscript.

**Competing interests:** The authors have declared that no competing interests exist.

the bites, infected salivary glands (SGs) release saliva containing viruses, which initiate human infection. As the tissue where transmitted viruses are produced, SG infection is a key determinant of transmission. To bridge the knowledge gap in vector-virus molecular interactions in SGs, we describe the transcriptome after DENV, ZIKV and CHIKV infection using RNA-sequencing and characterized the immune response in this tissue. Our study reveals the broad antiviral function of c-Jun N-terminal kinase (JNK) pathway against DENV, ZIKV and CHIKV in SGs. We further show that it is mediated by the complement system and apoptosis, identifying the mechanism. Our study adds the JNK pathway to the immune arsenal that can be harnessed to engineer refractory vectors.

## Introduction

In recent decades, dengue (DENV), Zika (ZIKV) and chikungunya (CHIKV) viruses have emerged as global public health issues, with over 50% of the world population at risk for infection [1]. DENV and ZIKV belong to the *Flavivirus* genus (Flaviviridae family), while CHIKV belongs to the *Alphavirus* genus (Togaviridae family). DENV infects an estimated 390 million people yearly, causing a wide range of clinical manifestations from mild fever to shock syndrome and fatal hemorrhage [2]. ZIKV recently emerged as an epidemic virus, infecting 1.5 million people over the past five years [3]. Although ZIKV infection is mostly asymptomatic, it can result in life-debilitating neurological disorders including Guillain-Barré syndrome in adults and microcephaly in prenatally-infected newborns [4]. CHIKV emerged as an epidemic virus in 2004, and has infected more than 6 million people [5]. It causes mild fever but can result in musculoskeletal inflammation, leading to long-term polyarthralgia. Amplified by urban growth, climate change and global travel, arboviral outbreaks are unlikely to recede in the near future [6].

DENV, ZIKV and CHIKV are primarily transmitted by *Aedes aegypti* mosquitoes. In the absence of efficient vaccines [7] and curative drugs [8], targeting this common vector is the best available strategy to control the spread of all three viruses. However, current vector control methods that rely on chemical insecticides are not effective in preventing outbreaks [9], partly due to insecticide resistance [10]. A novel strategy employing *Wolbachia* to reduce virus transmission by mosquitoes has been deployed as a trial in several countries [11]. However, its long-term efficacy may be compromised by converging bacteria and virus evolution [12]. Other promising approaches utilize genetic engineering technology to develop refractory vector populations [13]. Mosquito innate immunity can drastically reduce virus transmission, and characterization of mosquito immune pathways and mechanisms will aid in identifying gene candidates for transformation.

Immunity in midguts, the first organ to be infected following a blood meal, has been extensively studied by using transcriptomics. DENV and ZIKV activate the Toll, Immune Deficiency (IMD) and Janus Kinase (JAK)/Signal Transduction and Activators of Transcription (STAT) immune pathways [14–17]. Selective gene silencing studies have demonstrated the anti-DENV impact of Toll and JAK/STAT pathways. However, JAK/STAT transgenic activation was not effective in reducing ZIKV and CHIKV infection [13]. Immune effectors downstream of these pathways include antimicrobial peptides (AMPs) and thioester containing proteins (TEPs) [13–15,17–19]. AMPs such as Cecropin (Cec) and Defensin (Def) can have direct antiviral activity [20], while TEPs that belong to the complement system tag pathogens for lysis, phagocytosis, melanization [21,22] or induces AMPs [22]. Additionally, the RNA interference (RNAi) pathway cleaves viral RNA genomes [23]. However, its impact against

arbovirus infection is uncertain [24]. In *Drosophila melanogaster*, the Jun-N-terminal Kinase (JNK) pathway regulates a range of biological functions including immunity and apoptosis [25]. In *Anopheles gambiae*, the JNK pathway mediates an anti-malaria response through complement activation [26]. Currently, the impact of JNK pathway on arbovirus infection remains unexplored.

Following midgut invasion, arboviruses propagate to remaining tissues, including salivary glands (SGs), from where they are expectorated with saliva during subsequent bites. Despite the critical role of SGs in transmission, only three studies have examined DENV2-responsive differential gene expression in SGs. These studies revealed activation of the Toll and IMD pathways [20,27,28] and identified the anti-DENV2 functions of Cec [20], putative Cystatin and ankyrin-repeat proteins [28]. Here, we characterized the SG immune response to DENV2, ZIKV and CHIKV. We performed the first high throughput RNA-sequencing (RNA-seq) in infected SGs and observed differentially expressed genes (DEGs) related to immunity, apoptosis, blood-feeding and lipid metabolism. Using gene silencing, we discovered that upregulated components of the Toll and IMD pathways had variable effects against DENV2, ZIKV and CHIKV infections. However, for all three viruses, silencing of a JNK pathway upregulated component increased infection, and silencing of a negative regulator decreased infection in SGs. Further, we show that the JNK pathway is activated by all viruses and triggers a cooperative complement and apoptosis response in SGs. This work identifies and characterizes the JNK antiviral response that reduces DENV2, ZIKV and CHIKV infection in *A. aegypti* SGs.

## Results

### Transcriptome regulation by DENV2, ZIKV and CHIKV in SGs

SGs were collected at 14 days post oral infection (dpi) with DENV2 and ZIKV, and at seven dpi with CHIKV to account for variability between virus extrinsic incubation periods (EIP) [29,30]. To maximize the transcriptome impact, we chose virus titers that resulted in 100% infected SGs at the time of collection (S1A Fig). Differentially expressed genes (DEGs) were calculated with edgeR, DESeq2, and Cuffdiff 2, and showed little overlap among the algorithms (S2 Fig). To validate DEGs and select which software to use, we quantified the expression of 10 genes in a biological repeat with RT-qPCR and compared these values to the output from each algorithm. RNA-seq gene expression obtained with DESeq2 correlated best with RT-qPCR values (DENV2: $r^2 = 0.69$; ZIKV: $r^2 = 0.81$; CHIKV: $r^2 = 0.79$; S3 Fig), and only these DEGs are discussed.

The SG transcriptome was the most regulated by CHIKV infection (966 DEGs), followed by ZIKV (396) and DENV2 (202) (Fig 1A; S1 Table). Higher transcriptome regulation by CHIKV may stem from higher infection intensity as determined by viral genomic RNA (gRNA) copies (S1B Fig) and average percentage of RNAseq reads–e.g. DENV2, 11.16%; and ZIKV, 12.27%; CHIKV, 52.60%. Only 19 DEGs were common amongst the three virus infections (S1 Text), indicating a virus-specific transcriptome response. Comparison between DEGs from the current DENV2 infection and the three previous DENV2 studies with SGs collected at 14 dpi showed little overlap among them (S4 Fig) [20,27,28]. However, despite technical variations (e.g. mosquito colony, virus strain, transcriptomics technology) DEGs belonged to similar functional groups across the studies. We observed that immunity, apoptosis, blood-feeding and lipid metabolism related genes were highly regulated by DENV2, ZIKV and CHIKV (Fig 1B; DEGs related to apoptosis, blood-feeding and lipid metabolism are discussed in S1 Text).

A high proportion of DEGs were related to immunity with DENV2, ZIKV and CHIKV modulating 20, 118 and 86 immune genes, respectively (Fig 1B; S1 Table). Differential

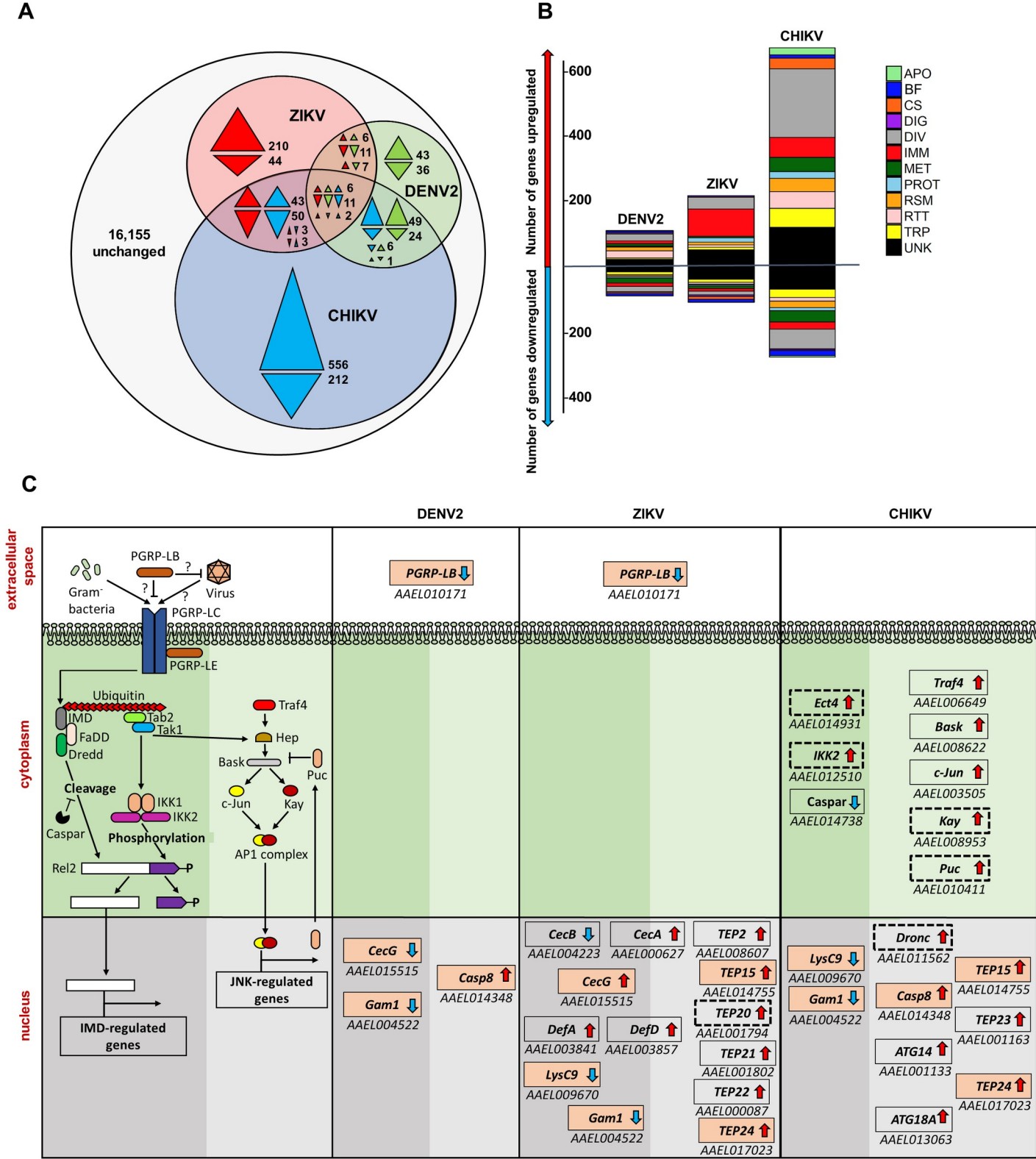

**Fig 1. DENV2, ZIKV and CHIKV induced transcriptome regulation in *A. aegypti* salivary glands. (A)** Venn diagram shows numbers of uniquely and commonly regulated differentially expressed genes (DEGs) in DENV2 (green), ZIKV (red) and CHIKV (blue) infected salivary glands. Colorless area shows total number of unchanged genes. Arrows indicate the direction of regulation for the corresponding color code. **(B)** Functional annotation of DEGs in DENV2, ZIKV or CHIKV infected salivary glands. APO, apoptosis; BF, blood feeding; CS, cytoskeleton and structure; DIG, digestion; DIV, diverse functions; IMM, immunity; MET,

metabolism; PROT, proteolysis; RSM, redox, stress and mitochondria; RTT, replication, transcription and translation; TRP, transport; UNK, unknown functions. (**C**) Transcriptomic regulation of the IMD and JNK pathways in DENV2, ZIKV and CHIKV infected salivary glands. First column represents a scheme of the JNK and IMD pathways partially, modified from Sim et al., [43]. In the following columns, boxes indicate DEGs with AAEL number below for DENV2, ZIKV and CHIKV. Arrows indicate the direction of regulation. Pink-filled boxes indicate genes regulated by more than one virus. Boxes with dotted line indicate DEGs selected for functional studies. Dark and light shaded green areas differentiate IMD from JNK pathways.

regulation of the RNAi, Toll, IMD, JAK/STAT and JNK pathways were observed (S5–S7 Fig). For the RNAi pathway, *Dicer-2* (*Dcr2)* was upregulated by all viruses, while *Argonaute-2* (*Ago2)* was upregulated by DENV2 only (S5 Fig). From the Toll pathway, only components upstream of the cytoplasmic cascade were regulated. We observed that ZIKV and CHIKV upregulated *Gram-negative binding protein A1* (*GNBPA1*) and downregulated *GNBPB6*, while ZIKV also upregulated *peptidoglycan recognition protein* S1 (*PGRPS1*) and CHIKV downregulated *GNBP2* (S6 Fig). Numerous serine proteases [e.g. CLIPs including Snake-likes (Snk-like) or Easter-likes (Est-like)] and serine protease inhibitors (Serpins) were upregulated by the viruses, except for *Snk-like* AAEL002273 and *CLIPB41* downregulation by CHIKV, and *SPE* and *CLIPB37* down-regulation by DENV. For the IMD pathway, we observed that DENV2 and ZIKV downregulated *PGRP-LB*, and that CHIKV downregulated *Caspar* and upregulated *IKK2* and *Ectoderm-expressed 4* (*Ect4*) (Fig 1C). For the Jak/STAT pathway, we only observed an upregulation of *SOCS36E* by CHIKV (S7 Fig). For the JNK pathway, CHIKV upregulated most of the JNK core pathway components (i.e. *Basket*, *c-Jun*, *Kayak* and *Puckered*) and *Traf4* (Fig 1C).

## Upregulated components from JNK but not from Toll and IMD pathway reduce DENV2, ZIKV and CHIKV

To determine how immune response influences DENV2, ZIKV and CHIKV multiplication in SGs, we silenced nine previously uncharacterized immune genes that were upregulated by at least one of the viruses (Table 1). They included four protease genes that initiate the Toll pathway (*CLIPB13A*, *CLIPB21*, one of the *Est-like*, one of the *Snk-like*) (S6 Fig), two genes from the cytoplasmic signaling of the IMD pathway (*IKK2* and *Ect4*) (Fig 1C), one transcription factor gene of the JNK pathway [*Kayak* (*Kay*)] (Fig 1C), and two putative immune genes—*Galectin-5* (*Gale5*) and a juvenile hormone induced gene (*JHI*). *Gale5* shows antiviral function against O'nyong-nyong in *A. gambiae* [31], while juvenile hormone treatment regulates immune gene expression in *D. melanogaster* [32] and *JHI* was upregulated by all three viruses in our study (Table 1; S1 Table). RNAi-mediated gene silencing efficacy ranged from 50–85% in SGs (S8A Fig) and did not affect mosquito survival (S9A Fig). Variation in gene silencing efficiency is common to SG studies [28]. To bypass the midgut barrier, we intrathoracically inoculated mosquitoes with a non-saturating inoculum of DENV2, ZIKV or CHIKV, resulting in 70–80% infected SGs (S10 Fig). This permitted the evaluation of an increase or decrease in infection upon gene silencing. At 10 days post inoculation, viral infection in SGs was measured using two parameters, infection rate and infection intensity. Infection rate was defined as the percentage of infected SGs out of 20 inoculated ones and represents dissemination of intrathoracically injected viruses into SGs. Infection intensity was calculated as gRNA copies per individual infected SGs and indicates virus replication. Of note, the two infection parameters do not reflect isolated biological phenomenon. For instance, a negative impact on infection intensity may lower infection rate due to virus clearance.

Silencing of most of the immune genes had a virus-specific effect on infection intensity and infection rate (Fig 2A–2C and Table 1). Silencing of *CLIPB13A* increased ZIKV infection rate, but decreased DENV2 and CHIKV infection rates. This suggests that *CLIPB13A* hinders ZIKV dissemination and facilitates DENV2 and CHIKV dissemination into SGs. However, it has a

**Table 1. Impact of immune response on DENV2, ZIKV and CHIKV infection in salivary glands.**

| Pathway | Gene | Virus | Induced[1] Log2(fold-change) | Function[2] | | | |
|---|---|---|---|---|---|---|---|
| | | | | Proviral | | Antiviral | |
| | | | | Dissemination | Replication | Dissemination | Replication |
| Toll | CLIPB13A (AAEL003243) | DENV2 | | [red] | | | |
| | | ZIKV | 0.7055 | | | [blue] | |
| | | CHIKV | 0.6353 | [red] | | | |
| | CLIPB21 (AAEL001084) | DENV2 | | | | | |
| | | ZIKV | 1.3290 | | | [red] | [blue] |
| | | CHIKV | 1.0705 | | | | |
| | Est-like (AAEL012775) | DENV2 | | | | | |
| | | ZIKV | | | | | |
| | | CHIKV | 1.1382 | [red] | | | |
| | Snk-like (AAEL002273) | DENV2 | 1.5291 | | | | |
| | | ZIKV | | | | | |
| | | CHIKV | 2.4517 | | | [blue] | |
| IMD | IKK2 (AAEL012510) | DENV2 | | | | | |
| | | ZIKV | | | [red] | [blue] | |
| | | CHIKV | 0.6699 | | | | |
| | Ect4 (AAEL014931) | DENV2 | | | | | [blue] |
| | | ZIKV | | | | | |
| | | CHIKV | 0.4973 | [red] | | | |
| JNK | Kayak (AAEL008953) | DENV2 | | | | | [blue] |
| | | ZIKV | | | | [blue] | [blue] |
| | | CHIKV | 0.5394 | | | [blue] | [blue] |
| Putative immune | JHI (AAEL000515) | DENV2 | 1.3759 | | | | |
| | | ZIKV | 0.7269 | | | | |
| | | CHIKV | 0.8029 | | | | |
| | Gale5 (AAEL003844) | DENV2 | 0.6647 | | | | |
| | | ZIKV | | [red] | [red] | | |
| | | CHIKV | 1.4059 | | | [blue] | |

[1]Induction in SGs as measured with RNA-seq.

[2]Determined by RNAi-mediated silencing studies in SGs.

minor role in regulating virus replication in SGs. *CLIPB21* silencing increased ZIKV infection rate but decreased its infection intensity. *Est-like* silencing decreased, whereas *Snk-like* silencing increased CHIKV infection rate. Overall, Toll pathway upregulated components showed mostly a virus-specific impact on dissemination into SGs (Fig 2A–2C and Table 1). Silencing of *IKK2* increased infection rate and decreased infection intensity for ZIKV. *Ect4* silencing increased DENV2 infection intensity and decreased CHIKV infection rate. Overall, IMD pathway response appears both proviral and antiviral (Fig 2A–2C; Table 1). *Gale5* silencing decreased both ZIKV infection intensity and infection rate, and enhanced CHIKV infection rate. Although *JHI* was upregulated by all viruses, it had no effect on any infections (Fig 2A–2C; Table 1). The silencing studies reflect a complex interaction between transcriptomic response and antiviral functions (Table 1) with the notable exception of Kay. *Kay* silencing increased infection rate to 100% for both ZIKV and CHIKV, and increased infection intensities by 8.6-, 6.75- and 17.65-fold for DENV2, ZIKV and CHIKV, respectively (Fig 2A–2C; Table 1). These results reveal a broad antiviral function of the JNK pathway in SGs. To

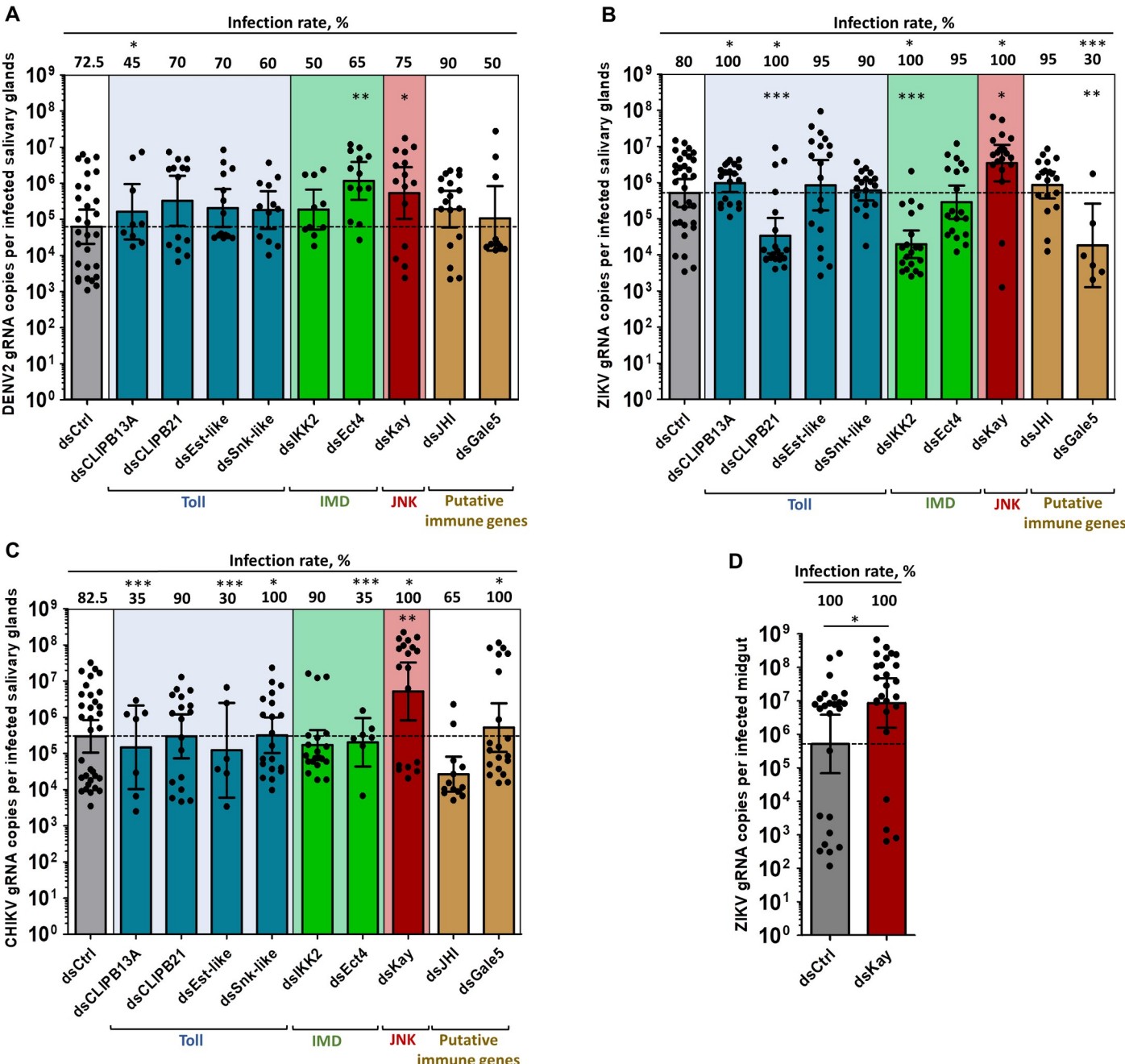

**Fig 2. Kayak depletion but not Toll or IMD component depletion increases salivary glands infection by DENV2, ZIKV and CHIKV, and midgut infection by ZIKV.** Four days post dsRNA injection, mosquitoes were infected by intrathoracic inoculation with DENV2, ZIKV or CHIKV, or by oral feeding with ZIKV. Viral genomic RNA (gRNA) was quantified at 10 days post inoculation in salivary glands and 7 days post oral infection in midguts. (**A-C**) Effect of immune-related gene silencing on gRNA copies and infection rate in salivary glands infected with (**A**) DENV2, (**B**) ZIKV, and (**C**) CHIKV. (**D**) Impact of *Kayak* silencing on gRNA number and infection rate in ZIKV-infected midgut. Bars show geometric mean ± 95% C.I. from 20 individual pairs of salivary glands or 25 individual midguts. Each dot represents one sample. ds*Ctrl*, dsRNA against LacZ; ds*CLIPB13A*, dsRNA against CLIP domain serine protease B13A; ds*CLIPB21*, dsRNA against CLIP domain serine protease B21; ds*Est-like*, dsRNA against Easter-like; ds*Snk-like*, dsRNA against Snake-like; ds*IKK2*, dsRNA against Inhibitor of nuclear factor kappa-B kinase; ds*Kay*, dsRNA against Kayak; ds*Ect4*, dsRNA against Ectoderm expressed-4; ds*JHI*, dsRNA against Juvenile hormone inducible; ds*Gale5*, dsRNA against Galectin 5. *, $p < 0.05$; **, $p < 0.01$; ***, $p < 0.001$, determined by post hoc Dunnett's with ds*Ctrl*, unpaired t-test or, for infection rate, Z-test.

determine whether JNK pathway also restricts virus in midguts, we orally infected *Kay*-silenced mosquitoes with ZIKV (S8B Fig). At seven dpi, while infection rate was already saturated at 100% in the control, infection intensity increased in midguts (Fig 2D). Overall, our functional studies discovered the antiviral and proviral roles of several immune-related genes and established that JNK pathway has a broad ubiquitous antiviral function.

## JNK pathway is induced by DENV2, ZIKV and CHIKV and reduces infection in SGs

SG transcriptomics showed that *Kay* was induced by CHIKV at seven dpi, but not by DENV2 or ZIKV at 14 dpi (Fig 1C). To test whether JNK pathway is activated by the three viruses, we monitored the kinetics of *Kay* expression in SGs after oral infection with either of the three viruses. *Kay* was similarly induced by DENV2, ZIKV and CHIKV at three and seven dpi, but not at 14 dpi (Fig 3A), corroborating the RNA-seq data. We quantified *Puckered (Puc)* expression, which is induced as a negative regulator [33]. Although not regulated at three and 14 dpi, *Puc* expression was increased at seven dpi for all three viruses (Fig 3B). We then tested the impact of *Puc* silencing (S8C Fig) in ZIKV-inoculated mosquitoes. *Puc* silencing did not alter mosquito survival (S9B Fig). Although SG infection rate was not affected, SG infection intensity was decreased by 8.85-fold at 7 days post inoculation (Fig 3C). Of note, this was achieved despite a moderate silencing efficiency of 35% (S8C Fig). Altogether, our data demonstrate that the JNK pathway is induced by DENV2, ZIKV and CHIKV at a time that corresponds to the onset of infection in SGs [30] and further support the JNK antiviral function.

## The JNK pathway antiviral response is mediated by TEP20 and Dronc in SGs

JNK pathway can regulate complement system [26], apoptosis [34] and autophagy [35]. To determine whether JNK pathway induces these functions in SGs, we monitored the impact of

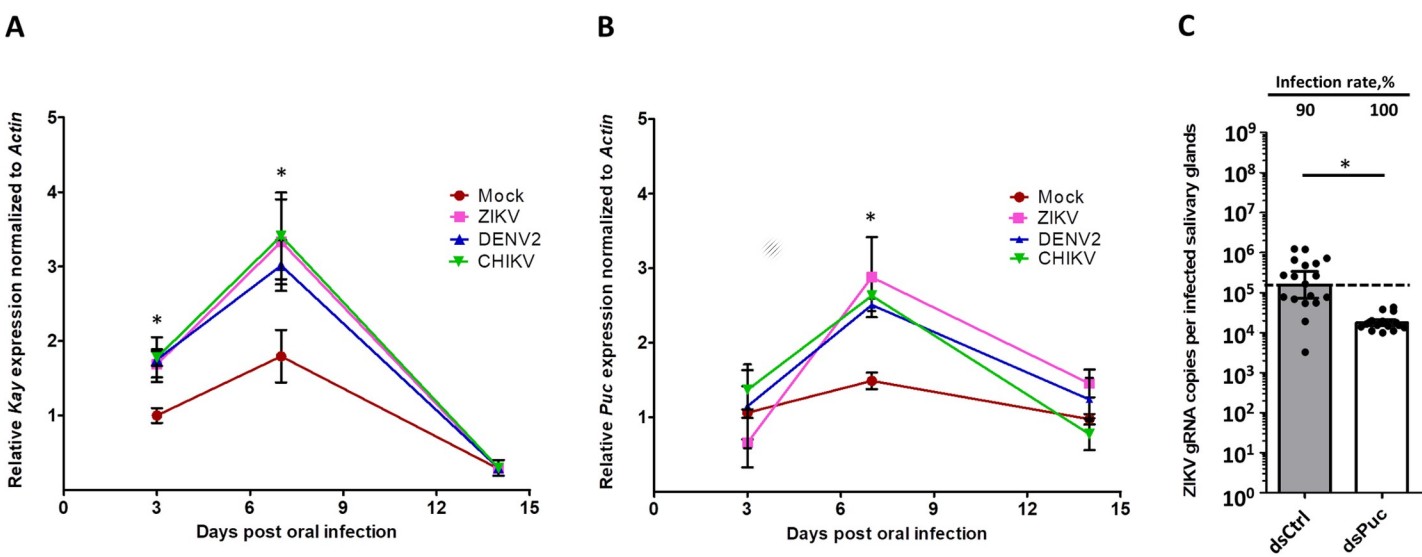

**Fig 3. *Kayak* and *Puckered* expressions are induced by DENV2, ZIKV and CHIKV infection, and Puckered depletion restricts ZIKV infection in salivary glands.** **(A)** *Kayak* (*Kay*) and **(B)** *Puckered* (*Puc*) expressions in salivary glands at 3, 7 and 14 days post oral infection with DENV2, ZIKV and CHIKV. Gene expression was quantified in pools of 10 salivary glands. *Actin* expression was used for normalization. Bars show arithmetic means ± s.e.m. from three biological replicates. **(C)** Effect of Puc depletion on infection in salivary glands at 7 days post ZIKV inoculation. Four days prior infection, mosquitoes were injected with dsRNA. Bars show geometric means ± 95% C.I. from 20 individual pair of salivary glands. Each dot represents one sample. ds*Ctrl*, dsRNA against LacZ, dsPuc, dsRNA against *Puc*. *, p < 0.05, as determined by Dunnett's test within time points with mock infection as control (A, B) or unpaired t-test (C).

*Kay* silencing on expressions of four TEPs (*TEP20*, *TEP24*, *TEP15* and *TEP2*), two pro-apoptotic (*Caspase8* and *Dronc*) and two autophagy-related (*ATG14* and *ATG18A*) genes at 10 dpi with ZIKV. All these genes were upregulated by infection in the transcriptomic data (Fig 1C; S1 Table). Control mosquitoes were injected with control dsRNA (dsCtrl). *Kay* silencing significantly reduced expressions of *TEP20* and *Dronc*, and moderately reduced *TEP15* and *TEP24* (Fig 4A). *TEP2* expression was increased and the two *ATGs* and *Caspase8* were unaffected. Based on these results we hypothesized that upon infection the JNK pathway induces the complement system through TEPs, and apoptosis through Dronc, but not Caspase8.

To test the antiviral function of TEP20 and Dronc in SGs, we challenged either *TEP20*- or *Dronc*-silenced mosquitoes (S8D Fig) by intrathoracically inoculating ZIKV. Because of the high sequence homology among TEPs, we ensured that dsRNA injected against *TEP20* specifically silenced *TEP20* and not *TEP2*, *TEP15*, *TEP22* and *TEP24*, which were regulated by virus infection in the RNA-seq analysis (S11 Fig). Similar to *Kay*-silencing, *TEP20*- and *Dronc*-silencing increased SG infection intensity by 21-fold to $2.6 \times 10^6$ gRNA and by 12-fold to $1.3 \times 10^6$ gRNA, respectively (Fig 4B). Since both complement and apoptosis can interact for cell clearance [36], we determined whether TEP20 and Dronc act in the same antiviral pathway by evaluating their synergistic effect when both genes were co-silenced (S8E Fig). We did not observe a clear difference in SG infection between *TEP20* and *Dronc* individual or co-silencing (Fig 4C and 4D). As the infection conditions did not saturate the infection intensity (higher inoculum resulted in $10^8$ gRNA per infected SG; S10 Fig), the lack of synergism when TEP20 and Dronc are co-silenced suggests that TEP20 and Dronc function in the same antiviral pathway. To confirm that TEP20 and Dronc mediate the JNK antiviral response in SGs, we induced JNK pathway by silencing of *Puc* and co-silenced *TEP20* or *Dronc* before ZIKV inoculation. Control mosquitoes were injected with the same quantity of dsCtrl or dsRNA against *Puc*. While infection intensity decreased upon *Puc* silencing, co-silencing of *Puc* with *TEP20* or *Dronc* restored viral gRNA copies to the level in dsCtrl-injected SGs (Fig 4D). These results demonstrate that JNK antiviral response is mediated by TEP20 and Dronc.

## TEP20 regulates AMPs and Dronc induces apoptosis in infected SGs

In *A. aegypti*, a macroglobulin complement-related factor (AaMCR) that belongs to the TEP family interacts with DENV2 through the scavenger receptor C (AaSR-C) and induces AMP expressions to reduce virus infection [22]. To test whether TEP20 regulates AMP expressions, we quantified expression of three AMPs (*CecA*, *DefA* and *DefD*) upon *TEP20* silencing (S8D Fig) in SGs at 10 days post ZIKV inoculation. These AMPs were upregulated by ZIKV in the RNA-seq transcriptome analysis (S6 Fig; S1 Table). TEP20 depletion significantly reduced the expression of *CecA* and *DefD*, while the expression of *DefA* remained unchanged (Fig 5A). TEPs that contain a thioester domain can directly bind to pathogens [37]. Protein sequence observation and alignment with AaMCR revealed the presence of a thioester motif (GCGEQ) in TEP20 (S12 Fig). These results indicate that TEP20 can interact to DENV2 to induce AMPs.

Dronc triggers apoptosis by cleaving effector caspases [38]. To support the role of Dronc in apoptosis in SGs, we depleted Dronc in SGs (S8D Fig) and quantified apoptotic cells at 10 days post ZIKV inoculation by using TUNEL assay. As compared to dsCtrl-injected mosquitoes, *Dronc* silencing significantly reduced the number of apoptotic cells (Fig 5A and 5B). Alternatively, TEP binding can mediate clearance of apoptotic bodies in mosquitoes, as observed by increased remaining apoptotic cells upon TEP1 depletion [39]. To determine the role of TEP20 in apoptotic clearance, we quantified apoptotic cells in *TEP20*-silenced SGs (S8D Fig) dissected at 10 days post ZIKV inoculation. We did not observe any impact of TEP20 depletion

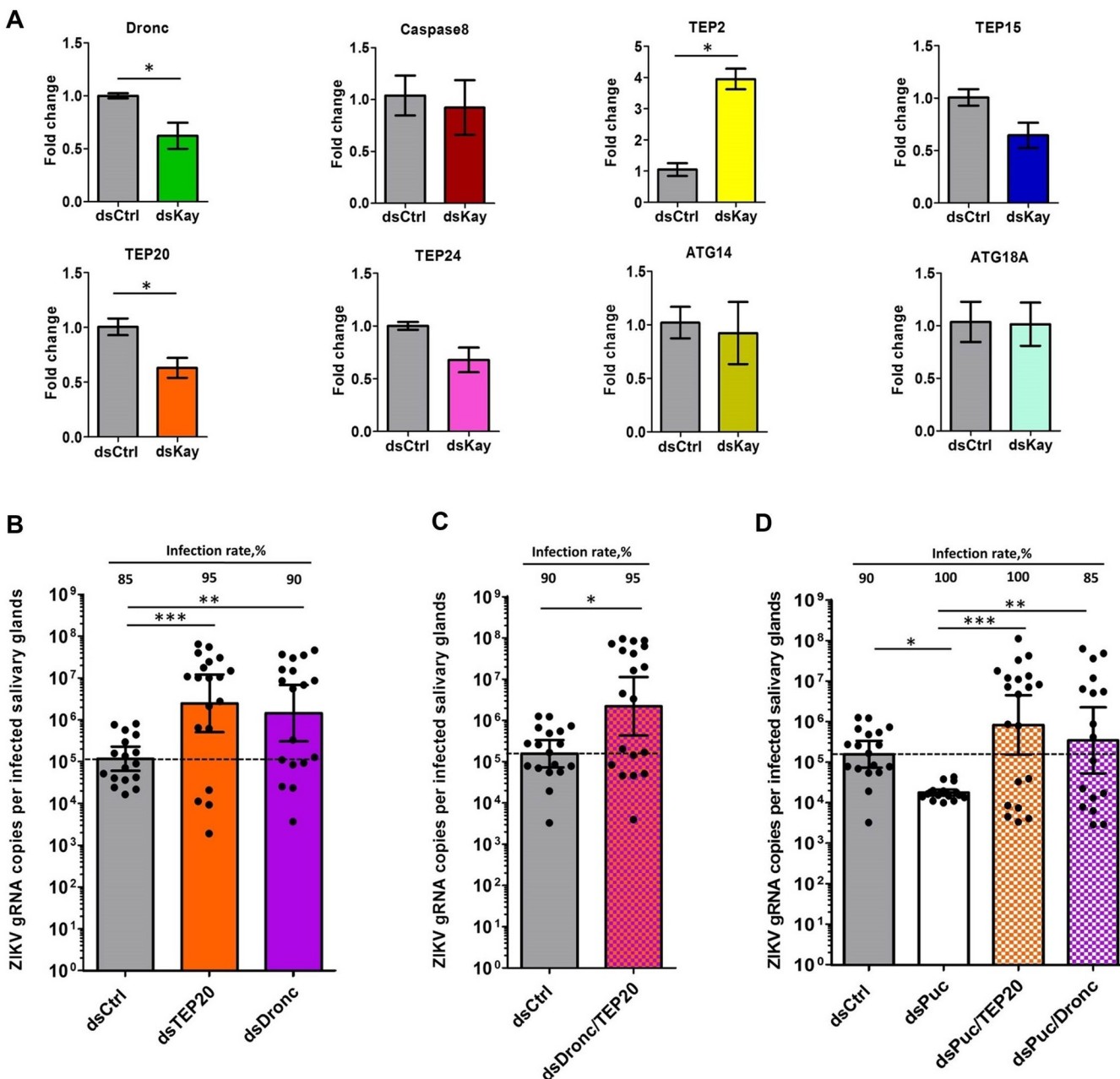

**Fig 4. Virus-induced JNK pathway activates an antiviral response through TEP20 and Dronc. (A)** Impact of Kayak depletion on expression of genes related to complement system (*TEP20*, *TEP24*, *TEP15* and *TEP2*), apoptosis (*Caspase8* and *Dronc*) and autophagy (*ATG14* and *ATG18A*) at 10 days post ZIKV oral infection. Gene expression was quantified in pools of 10 salivary glands. *Actin* expression was used for normalization. Bars show arithmetic means ± s.e.m. from three biological replicates. **(B-D)** Effect of depletion of **(B)** Dronc or TEP20 alone, **(C)** Dronc and TEP20 simultaneously, and **(D)** Puc alone, or Puc and Dronc simultaneously, or Puc and TEP20 simultaneously on salivary gland infection at 7 days post ZIKV inoculation. Four days prior infection, mosquitoes were injected with dsRNA. Bars show geometric means ± 95% C.I. from 20 individual salivary glands. Each dot represents one sample. *TEP*, Thioester-containing protein; *ATG*, Autophagy related gene; ds*Kay*, dsRNA against Kayak; ds*Puc*, dsRNA against Puckered; ds*Ctrl*, dsRNA against LacZ; dsTEP20, dsRNA against *TEP20;* dsDronc, dsRNA against *Dronc*. *, p < 0.05; **, p < 0.01; ***, p < 0.001, as determined by unpaired t-test (A) or Dunnett's test with dsCtrl as control (B-D).

on apoptotic cells, whereas simultaneous depletion of Dronc and TEP20 reduced apoptotic bodies (Fig 5A and 5B). These results indicate that apoptosis in infected SGs depends on Dronc but not TEP20.

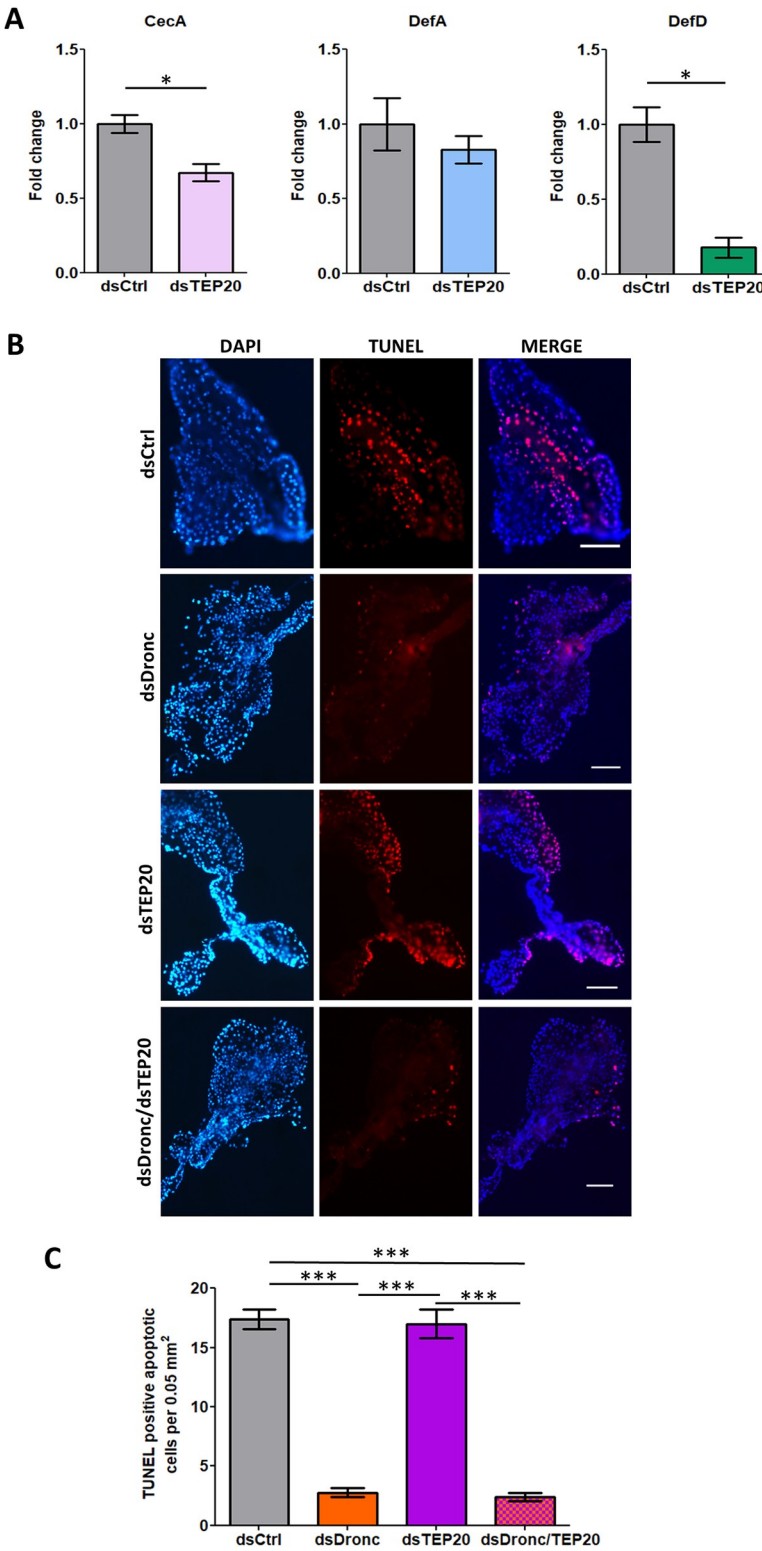

**Fig 5. Functions of TEP20 and Dronc in SGs.** DsRNA-injected mosquitoes were inoculated with ZIKV and analyzed 10 days later. Controls were injected with dsCtrl. **(A)** Impact of TEP20 depletion on expression of the AMPs: *CecA*, *DefA* and *DefD*. Gene expression was quantified in pools of 10 salivary glands. *Actin* expression was used for normalization. Bars show arithmetic means ± s.e.m. from three biological replicates. **(B)** Apoptotic staining in SGs upon depletion of Dronc (dsDronc), TEP20 (dsTEP20) and both simultaneously (dsDronc/dsTEP20). Apoptotic cells

were detected via TUNEL staining (red) and nuclei were stained using DAPI (blue). Scale bar, 200 μm. (**C**) Number of apoptotic cells in salivary glands upon depletion of Dronc, TEP20 and both. Cells were counted on twelve 0.05 mm$^2$ areas observed from two SGs per condition. ds*Ctrl*, dsRNA against LacZ; dsTEP20, dsRNA against *TEP20;* dsDronc, dsRNA against *Dronc; CecA*, Cecropin A; *DefA*; Defensin A, *DefD*, Defensin D. $^{***}$, p < 0.001, as determined by Tukey's post-hoc test.

## Discussion

Dengue, Zika and chikungunya are widespread mosquito-borne diseases primarily transmitted by *A. aegypti*. Disease mitigation through insecticide-based vector control has mostly remained ineffective to prevent outbreaks [7]. Currently, insecticide-free vector-control strategies are being extensively investigated. The improved understanding of the vector-virus interaction at the molecular level proposed here paves the way to manipulate mosquito biology to make them refractory towards arboviruses. This study discovers the antiviral function of the JNK pathway in *A. aegypti* SGs against three major arboviruses, DENV2, ZIKV and CHIKV, belonging to two virus families. Further, we determine that the antiviral response is mediated through induction of complement system and apoptosis.

Several studies have functionally characterized the immune response in *A. aegypti* midgut, the first barrier to infection, determining the antiviral function of Toll, IMD and JAK/STAT pathways [14,15,27,40]. However, only a couple of studies tested the impact of SG (exit barrier) immune response [20,28]. Luplertlop et al.[20] used Digital Gene Expression tag profiling to quantify the impact of DENV2 infection in SGs. They reported overexpression of Toll pathway components, but did not test their functions. Instead, they characterized the most abundant protein, an AMP from the Cec family (AAEL000598), and revealed its broad antiviral function *in vitro*. This Cec was not regulated in our transcriptomic analysis, although two other Cec (CecA and G) were differentially modulated by the different viruses. Sim et al.[28] used microarrays to identify SG-specific transcripts as compared to carcasses. An enrichment in immune-related genes suggested the ability to mount an immune response in SGs. Further, in the same study, they determined the transcriptome upon DENV2 infection. Similar to our data, they reveal a high regulation of serine proteases that could initiate the different immune pathways or play a role in blood feeding when secreted. While they did not functionally test the immune pathways, Sim et al. [28] revealed the antiviral and proviral functions of three DENV2-upregulated genes. This supports the complex interaction between gene regulation and function that we also observed.

In our study, we functionally tested for the first time the impact of upregulated components from Toll, IMD and JNK pathways on DENV2, ZIKV and CHIKV in SGs. The Toll pathway is triggered when extracellular pattern recognition receptors (PRR) (e.g. PGRP and GNBP) bind to pathogen-derived ligands and activate a proteolytic cascade that leads to activation of pro-Spätzle to Spätzle by Spätzle processing enzyme (SPE) (S6 Fig) [41]. Spätzle binding to transmembrane receptor Toll induces a cytoplasmic cascade that leads to nuclear translocation of NF-κB transcription factor Rel1a to initiate effector gene transcriptions. We only observed a regulation of pattern recognition receptors (PRRs) and pre-cytoplasmic proteases. Of note, downregulation of the SPE protease by DENV is reminiscent of downregulation of Toll pathway components by subgenomic flaviviral RNA (sfRNA) in SGs [42]. The IMD pathway, similarly to Toll, is triggered by microbial ligand recognition by PRRs, which activate PGRP-LC transmembrane protein (Fig 1C). In the ensuing cytoplasmic cascade, NF-κB transcription factor Rel2 is phosphorylated by IKK2 and cleaved by DREDD to induce its nuclear translocation. IMD pathway is repressed by Caspar [43], Ect4 [44] and PGRP-LB [45]. IMD may have been activated upon DENV and ZIKV infection through PGRP-LB downregulation and upon

CHIKV infection through Caspar downregulation and IKK2 upregulation, while Ect4 upregulation by CHIKV may have controlled the activation. The JAK/STAT pathway is triggered by binding of Unpaired (Upd) to transmembrane protein Domeless (Dome), which dimerizes and phosphorylates Hopscotch (Hop) (S7 Fig) [43]. The activated Dome/Hop complex phosphorylates STATs, which upon dimerization translocate to nucleus and initiate transcription. The phosphorylation of STATs is controlled by Suppressor of Cytokine Signaling (SOCS) family of inhibitors [46]. SOCS36E upregulation upon CHIKV infection suggests pathway inhibition. The JNK pathway is triggered by Hemipterous (Hep) phosphorylation through tumor necrosis-associated factor 4 (Traf4) or Tak1, the latter being an IMD pathway component (Fig 1C). Hep activates Basket (Bask) that phosphorylates c-Jun and Kay transcription factors. Activated c-Jun and Kay dimerize to form the activator protein-1 (AP-1) complex, which translocates to nucleus and induces transcription. *Puc* transcription is induced by the JNK pathway and represses Bask [33], acting as a feedback loop. Upregulation of most of the core components by CHIKV infection strongly indicates JNK pathway activation, while Traf4 upregulation suggests an IMD-independent activation. The lack of impact against DENV2, ZIKV and CHIKV when silencing the upregulated components of Toll and IMD pathways may not reflect the antiviral capabilities of these pathways. Indeed, it is possible that a complete shutdown of the signaling cascades increases virus infection. Importantly, however, we discovered the antiviral impact of the JNK pathway against DENV2, ZIKV and CHIKV in SGs.

JNK pathway activation by DENV2, ZIKV and CHIKV in SGs occurred early during infection at 3 dpi. This time corresponds to the onset of SG infection [30]. Such an early induction may affect virus dissemination to SGs as reflected in ZIKV and CHIKV infection rates. The JNK pathway can be induced either through IMD-mediated pathogen recognition or oxidative stress. Tak1 activates the IMD and JNK pathways through IKK2 and Hep, respectively [47]. Since Tak1-mediated JNK activation is transient, lasting less than one hour [48], the observed JNK activation (lasting at least 3–7 dpi) is probably due to an IMD-independent activation. Upon oxidative stress, the JNK pathway is induced through p53 upregulation of *Traf4* that then phosphorylates Hep [49–51]. Activated JNK pathway then induces other oxidative stress-associated genes, such as *FoxO* and *κ-B Ras* (AAEL003817) [52]. In our transcriptomic data, we reported the upregulation of several oxidative stress-associated genes upon infection including *p53*, *Traf4*, *FoxO*, *CYPs* and *κ-B Ras* (S1 Table). These results suggest that JNK pathway is activated in SGs as a result of infection-induced oxidative stress.

Using *in vivo* functional studies, we showed that the JNK antiviral function depends on complement system and apoptosis inductions. The complement system is activated when TEPs bind to pathogens and trigger lysis, phagocytosis [37] or even AMP production [22,53]. TEPs such as TEP15 and AaMCR can restrict DENV in mosquitoes [22,54], although TEP22 does not [13]. In the current study, we show that TEP20 regulates expression of CecA and DefD in SGs. While DefD antiviral function has not been studied, CecA depletion increased DENV infection [19]. The data suggests that TEP20 antiviral function is at least partially mediated through AMP regulation. Apoptosis was previously reported in SGs infected with different flaviviruses and alphaviruses [55–57]. Apoptosis antiviral function in mosquitoes is supported by two observations. Firstly, an arbovirus with an apoptosis-inducing transgene was selected out during mosquito infection [58], and secondly, there is an association between the ability to induce apoptosis and colony refractoriness in different virus-mosquito systems [59–61]. These indicate that apoptosis can define vector competence, emphasizing its importance as a target to control transmission. In our study, we reported the antiviral function of the apoptotic activator Dronc. Furthermore, we tested whether TEP20 and Dronc act in the same pathway. A lack of synergistic effect when both TEP20 and Dronc were silenced suggests that complement system and apoptosis function in the same cascade to reduce virus infection.

Binding of complement components can mediate apoptotic cells clearance in mammals [36] and in testes of *A. gambiae* mosquitoes [39]. However, we did not observe any impact of TEP20 depletion on apoptotic cell number in infected SGs. Alternatively, apoptosis-triggered nitration of virus surface could be required to direct thioester-mediated TEP binding, as in *Anopheles* mosquitoes infected with parasites [62,63]. While co-regulation of Dronc and TEP20 by the JNK pathway and absence of any synergistic effect on virus infection upon their silencing strongly suggest that they function in the same pathway, the exact interaction among them is yet to be determined.

A recent study established a negative association between the presence of efficient *Plasmodium*-killing immune response in mosquitoes and epidemics in Africa, confirming the long-suspected impact of mosquito immunity on epidemiology for arthropod-borne diseases [64]. Upon close inspection of field-derived *A. aegypti* colony transcriptomes [15], we found that DENV2-refractory colonies expressed a higher level of *Kay*, c-*Jun* and *TEP20*. This suggests that variation in vector competence among these colonies is partially related to JNK pathway. Consistent with our study, this points to a role of JNK pathway in determining the arbovirus transmission dynamics in the field. Because JNK pathway is sensitive to various microbes [65], differential activation by distinct microbes present in natural habitats may also represent a trigger that influences transmission.

Development of transgenic refractory mosquitoes have gained prominence in preventing arboviral transmission [66]. Engineered overexpression of a JAK/STAT activator in mosquitoes reduced DENV2 propagation and established its proof-of-concept [13]. However, JAK/STAT is ineffective against ZIKV and CHIKV. To our knowledge, no promising candidates have been identified to be antiviral against DENV2, ZIKV and CHIKV. In this context, our work reveals that the JNK pathway components could be harnessed to develop effective transmission blocking tools against a broad range of arboviruses.

## Material and methods

### Mosquitoes

*Aedes aegypti* mosquitoes were collected in Singapore in 2010. Eggs were hatched in MilliQ water and larvae were fed with a mixture of fish food (TetraMin fish flakes), yeast and liver powder (MP Biomedicals). Adults were reared in cages (Bioquip) supplemented with 10% sucrose and water. The insectary was held at 28˚C and 50% relative humidity with a 12h:12h light:dark cycle.

### Virus isolates

The dengue virus serotype 2 PVP110 (DENV2) was isolated from an EDEN cohort patient in Singapore in 2008 [67]. The Zika virus Paraiba_01/2015 (ZIKV) was isolated from a febrile female in the state of Paraiba, Brazil in 2015 [68]. The chikungunya virus SGP011 (CHIKV) was isolated from a patient at the National University Hospital in Singapore [69]. DENV2 and ZIKV isolates were propagated in C6/36 and CHIKV in Vero cell line. Virus stocks were titered with BHK-21 cell plaque assay as previously described [70], aliquoted and stored at -80˚C.

### Oral infection

Three-to-five day-old female mosquitoes were starved for 24 h and offered an infectious blood meal containing 40% volume of washed erythrocytes from serum pathogen free (SPF) pig's blood (PWG Genetics), 5% 10 mM ATP (Thermo Scientific), 5% human serum (Sigma) and 50% virus solution in RPMI media (Gibco). Mosquitoes were left to feed for 1.5 h using

Hemotek membrane feeder system (Discovery Workshops) covered with porcine intestine membrane (sausage casing). The relative long feeding was required to warm the blood meal previously kept in the refrigerator to 37˚C to maximize feeding. The virus titers in blood meals were $2 \times 10^7$ pfu/ml for DENV2, $6 \times 10^6$ pfu/ml for ZIKV, and $1.5 \times 10^8$ pfu/ml for CHIKV, and validated in plaque assay using BHK-21 cells. Control mosquitoes were fed with the same blood meal composition except for virus. Fully engorged females were selected and kept in a cage with *ad libitum* access to a 10% sucrose solution in an incubation chamber with conditions similar to insect rearing.

### Virus inoculation

Mosquitoes were cold anesthetized and intrathoracically inoculated with either DENV2, ZIKV or CHIKV using Nanoject-II (Drummond). The same volume of RPMI media (ThermoFisher Scientific) was injected as control. Functional studies were conducted by inoculating 0.1 pfu of DENV2, 0.01 pfu of ZIKV and 0.2 pfu of CHIKV.

### SG collection, library preparation and RNA-sequencing

SGs from DENV2- and ZIKV-orally infected mosquitoes were dissected at 14 dpi. Those from CHIKV-orally infected mosquitoes were dissected at 7 dpi. Controls for CHIKV were dissected at 7 days post feeding on a non-infectious blood meal, and at 14 days post feeding for DENV2 and ZIKV. Inoculum resulted in 100% infected SGs (S1 Fig). Fifty pairs of SGs per condition were homogenized using a bead mill homogenizer (FastPrep-24, MP Biomedicals). Total RNA was extracted using E.Z.N.A Total RNA kit I (OMEGA Bio-Tek). RNA-seq libraries were prepared using True-Seq Stranded Total RNA with Ribo-Zero Gold kit (Illumina), according to manufacturer's instructions. Following quantification by RT-qPCR using KAPA Library Quantification Kit (KAPA Biosystems), libraries were pooled in equimolar concentrations for cluster generation on cBOT system (Illumina) and sequenced (150 bp pair-end) on a HiSeq 3000 instrument (Illumina) at the Duke-NUS Genome Biology Facility, according to manufacturer's protocols. Two repeats per condition were processed. The need for more than two repeats was mitigated by using a large number of tissues from different mosquitoes in each repeat. Raw sequencing reads from RNA-seq libraries are available online under NCBI accessions: SRR8921123-8921132.

### RNA-seq data processing and identification of differentially expressed genes (DEGs)

Reads were quality checked with FastQC (www.bioinformatics.babraham.ac.uk) to confirm that adapter sequences and low-quality reads (Phred+33 score < 20) had been removed. The reads were then aligned against the *A. aegypti* genome [VectorBase [71]] AaegL3.3) using TopHat v2.1.0 [72] with parameters–N 6 –read-gap-length 6 –read-edit-dist. 6 set to account for regional genetic variation between the RNA-seq and genome. SAM tools v0.1.19 [73] and HTSeq v0.6.0 [74] were used to format and produce count files for gene expression analysis. DEGs were identified using DESeq2 [75] with at least 1.4-fold change between control and infected conditions, an adjusted False Discovery Rate (FDR) of 0.05. edgeR [76] and Cuffdiff 2 [77] (following the Cufflinks pipeline [78] were also used to identify DEGs following the same criteria.

### Gene annotations

DEGs were annotated through several pipelines. Predicted gene protein sequences were searched against the NCBI nr database (accessed July 2017) (BLASTp; e-value 1.0E-5 and word

size 3), VectorBase [71] *Aedes* peptide database (downloaded July 2017) with BLAST+ [79] (BLASTp; e-value 1.0E-5 threshold and word size 2), and FlyBase [80] *D. melanogaster* peptide database (downloaded August 2017) with BLAST+ (BLASTp; e-value 1.0E-5 threshold and word size 2) for identification of orthologues. Gene ontology terms were assigned in BLAST2GO [81] (program default parameters). Signal peptides were identified using SignalP v4.1 [82]. Functional annotations were also assigned based on literature.

## Double-stranded mediated RNAi

Mosquito cDNA was used to amplify dsRNA targets with gene specific primers tagged with T7 promoter as detailed in S2 Table. The amplified products were *in vitro* transcribed with T7 Scribe kit (Cellscript). dsRNAs were annealed by heating to 95˚C and slowly cooling down to 4˚C using a thermocycler. Three to five-day-old adult female mosquitoes were cold-anesthetized and intrathoracically injected with 2 or 4 μg of dsRNA using Nanoject II. The same quantity of dsRNA against the bacterial gene *LacZ* was injected as control (dsCtrl). Validation of gene silencing was conducted 4 days post injection by pooling 10 SGs or 5 midguts.

## Gene expression quantification using RT-qPCR

Total RNA was extracted from 10 SGs or 5 midguts using E.Z.N.A. Total RNA kit I, DNAse treated using Turbo DNA-free kit (Thermo Fisher Scientific), and reverse transcribed using iScript cDNA synthesis kit (Biorad). Gene expression was quantified using qPCR with Sensi-Fast Sybr no-rox kit (Bioline) and gene specific primers detailed in S2–S4 Tables. *Actin* expression was used for normalization. The reactions were performed using the following cycle conditions: an initial 95˚C for 10 min, followed by 40 cycles of 95˚C for 5 s, 60˚C for 20 s and ending with a melting curve analysis. The delta delta method was used to calculate relative fold changes.

## Quantification of virus genome RNA (gRNA) copies using RT-qPCR

Individual pairs of either SGs or midguts were homogenized with a bead Mill Homogenizer in 350 μl of TRK lysis buffer (E.Z. N. A Total RNA kit I). Total RNA was extracted using the RNA extraction kit and reverse-transcribed using iScript cDNA synthesis kit. DENV2 gRNA copies were quantified by RT-qPCR using i-Taq one step universal probes kit (BioRad) and ZIKV and CHIKV gRNA copies with i-Taq one step universal sybr kit (BioRad) with primers detailed in S5 Table. Amplification was run on CFX96 Touch Real-Time PCR Detection System (BioRad) with the following thermal profile: 50˚C for 10 min, 95˚C for 1 min, followed by 40 cycles at 95 ˚C for 10 s, 60 ˚C for 15 s. A melt-curve analysis was added for Sybr qPCR.

Absolute quantification of gRNA was obtained by generating a standard curve for each virus target. Viral cDNA was used to amplify qPCR target using qPCR primers with T7-tagged forward primer. RNA fragments were generated with T7-Scribe kit and RNA concentration calculated by Nanodrop was used to estimate concentration of RNA fragments. Ten-time serial dilutions were quantified with RT-qPCR and used to generate absolute standard equations. Three standard dilutions were used in each subsequent RT-qPCR plate to adjust for inter-plate variation.

## Apoptotic cell staining

SGs were dissected at 10 days post ZIKV inoculation, dried on a SuperFrost Plus slide (ThermoFisher Scientific), fixed with 2% paraformaldehyde for 30min, permeabilized in 0.5% Triton X-100 (Sigma) for 15min at room temperature (RT) and blocked with 0.5% Triton X-100

and 1% BSA (Sigma) for 15 min at RT. After two PBS washes, TUNEL staining was performed with ApopTag Peroxidase In Situ Apoptosis Detection Kit (Merck). Briefly, the SGs were incubated in equilibration buffer for 20 sec at RT. After removing excess liquid, the SGs were incubated for 40min with TdT enzyme in a humidified chamber at 37°C. The samples were next incubated with the anti-digoxigenin conjugate in dark for 10min at RT, and later washed few times with PBS. Finally, the tissues were mounted with ProLong Gold antifade mountant that contains DAPI (Invitrogen). Pictures were taken with a fluorescence microscope (Nikon Eclipse 80i). The apoptotic cells were counted on twelve $0.05mm^2$ areas on images taken from two SGs per condition.

## Data analyses and software used

In figures, the geometric means of viral loads were shown. The geometric mean indicates the central tendency of a set of numbers by using the product of their values. Mathematically, the geometric mean calculates the arithmetic mean of a log-transformed data. Since viral loads followed a logarithmic distribution as determined by D'Agostina-Pearson Omnibus test (K2) after logarithmic transformation, we chose geometric mean to represent their central tendency.

One-way ANOVA and post-hoc Dunnett's test or unpaired T-test were used to test differences in gene expression and log10-transformed gRNA copies. Normal distribution of log-transformed viral loads (gRNA) was confirmed by K2 test. These analyses were done using GraphPad Prism 5. Z-score was used to test differences in infection rate with www.socscistatistics.com/tests/ztest/. Standard error for sample proportion was calculated with www.easycalculation.com.

## Supporting information

**S1 Text. Analysis of DEGs commonly regulated by DENV2, ZIKV and CHIKV, DEGs related to immune effectors, apoptosis, blood-feeding and lipid metabolism.**
(DOCX)

**S1 Table. Fold changes and functional annotations for DEGs in SGs upon infection with DENV2, ZIKV and CHIKV.** (Excel sheet).
(XLSX)

**S2 Table. Primer sequences for candidate gene RNAi silencing and RT-qPCR quantification**
(DOCX)

**S3 Table. Primer sequences for DEG validation**
(DOCX)

**S4 Table. Primer sequences for JNK pathway-controlled gene expressions and corresponding RNAi silencing**
(DOCX)

**S5 Table. Primer sequences for virus quantification**
(DOCX)

**S1 Fig. Infection level in salivary glands from mosquitoes orally fed with different inocula of DENV2, ZIKV and CHIKV.** Mosquitoes were orally infected with a blood meal containing $10^5$ to $10^8$ pfu per ml of either DENV2, ZIKV or CHIKV. At 14 days post infection with DENV2 and ZIKV, and 7 days post infection with CHIKV, 20 salivary glands were dissected

and virus was quantified using RT-qPCR. **(A)** Infection rate as measured by the percentage of infected SGs. Bars show percentage ± standard error. **(B)** Infection intensity as measured by gRNA copies per infected salivary glands. Inocula presented were the one selected for RNAseq. Bars show geometric means ± 95% C.I.
(TIF)

**S2 Fig. Comparisons of Differentially Expressed Genes (DEGs) obtained with edgeR, DESeq2 and Cuffdiff 2. (A-C)** Venn diagrams presenting overlaps in up- and downregulated DEGs between edgeR, DESeq2 and Cuffdiff 2 in salivary glands infected with **(A)** DENV2, **(B)** ZIKV and **(C)** CHIKV.
(TIF)

**S3 Fig. Correlations between RNA-seq and RT-qPCR gene expressions. (A-C)** The expression of 10 genes in **(A)** DENV2, **(B)** ZIKV or **(C)** CHIKV infected salivary glands was quantified with RT-qPCR and correlated to their respective fold change determined from DESeq2, edgeR or CuffDiff 2 outputs. Log2 Fold-Change (log2FC) is displayed on axes. Three replicates of 10 salivary glands were used for RT-qPCR. RT-qPCR and RNA-seq samples were collected from different biological repeats. $r^2$ indicates Pearson correlation for gene expressions between the two methods.
(TIF)

**S4 Fig. Comparisons of Differentially Expressed Genes (DEGs) in DENV2 infected salivary glands between our dataset and three previous studies.** Venn diagram showing common and different DEGs in (i) Singapore *A. aegypti* colony orally infected with PVP110 virus (this study), (ii) Rockefeller/UGAL *A. aegypti* colony orally infected with New Guinea C virus [28], (iii) Liverpool *A. aegypti* colony orally infected with 16681 virus [20], (iv) Chetumal *A. aegypti* colony orally infected with Jam1409 virus [27].
(TIF)

**S5 Fig. Transcriptomic regulation of the RNAi pathway in DENV2, ZIKV and CHIKV infected salivary glands.** Boxes indicate differentially expressed genes (DEGs) with AAEL number below. Arrows indicate the direction of regulation. Pink boxes indicate DEGs by more than one virus.
(TIF)

**S6 Fig. Transcriptomic regulation of the Toll pathway in DENV2, ZIKV and CHIKV infected salivary glands.** Boxes indicate differentially expressed genes (DEGs) with AAEL number below. Arrows indicate the direction of regulation. Pink boxes indicate DEGs by more than one virus. Dotted boxes show genes selected for functional studies.
(TIF)

**S7 Fig. Transcriptomic regulation of the JAK/STAT pathway in DENV2, ZIKV and CHIKV infected salivary glands.** Boxes indicate differentially expressed genes (DEGs) with AAEL number below. Arrows indicate the direction of regulation. Pink boxes indicate DEGs by more than one virus.
(TIF)

**S8 Fig. Silencing efficiencies in salivary glands and midgut.** Each mosquito was injected with dsRNA against the candidate gene. Same quantity of dsCtrl was injected as control. Four days later, mRNA was quantified using RT-qPCR in pools of 10 salivary glands or 5 midguts. *Actin* expression was used for normalization. Corresponding gene expression after injection with **(A)** 2 µg of dsCLIPB13A, dsCLIPB21, dsEst-like, dsSnk-like, dsIKK2, dsEct4, dsKay,

dsJHI or dsGale5 in salivary glands; (**B**) 2 μg of dsKay in midgut; (**C**) 2 μg of dsTEP20 or dsDronc in salivary glands; (**D**) 4 μg of equal amount of dsTEP20 and dsDronc in salivary glands; (**E**) 2 μg of dsPuc in salivary glands; and (**F**) 4 μg of dsPuc, or equal amount of dsTEP20 and dsDronc in salivary glands. Bars show means ± s.e.m. from three repeats. dsRNA target: ds*Ctrl*, LacZ; ds*CLIPB13A*, CLIP domain serine protease 13A; ds*CLIPB21*, CLIP domain serine protease B21; ds*Est-like*, Easter-like; ds*Snk-like*, Snake-like; ds*IKK2*, Inhibitor of nuclear factor kappa-B kinase; ds*Kay*, Kayak; ds*Puc*, Puckered; ds*Ect4*, Ectoderm expressed-4; ds*JHI*, Juvenile hormone inducible; ds*Gale5*, Galectin 5; ds*Dronc*, Dronc; ds*TEP20*, Thioester containing protein 20. \*, p < 0.05; \*\*, p < 0.01, as determined by unpaired t-test.
(TIF)

**S9 Fig. Mosquito survival upon gene silencing.** Each mosquito was injected with 2μg of dsRNA against the candidate gene or control dsRNA (ds*Ctrl*). Mosquito survival at 4 days post dsRNA injection against (**A**) the candidate immune genes and (**B**) *Puckered* (*Puc*). Bars show percentage ± standard error. N, number of dsRNA-injected mosquitoes.
(TIF)

**S10 Fig. Quantification of salivary gland infection after inoculation with different inocula of DENV2, ZIKV and CHIKV.** Four days after injection with dsRNA control (ds*Ctrl*), mosquitoes were inoculated with different inoculum doses (plaque forming unit, pfu) of either DENV2, ZIKV or CHIKV. Ten days later, viral genomic RNA (gRNA) was quantified in 20 salivary glands. gRNA copies and infection rate in salivary glands from mosquitoes inoculated with (**A**) DENV2, (**B**) ZIKV and (**C**) CHIKV. Each dot represents one pair of salivary glands. Bars show geometric means ± 96% C.I.
(TIF)

**S11 Fig. Silencing specificity of dsTEP20 in salivary glands.** Each mosquito was injected with dsRNA against TEP20 (dsTEP20). Same quantity of dsCtrl was injected as control. Four days later, mRNA was quantified using RT-qPCR in pools of 10 salivary glands. *Actin* expression was used for normalization. Salivary gland gene expression for TEP20, TEP2, TEP15, TEP22 and TEP24. Bars show arithmetic means ± s.e.m. from three repeats. \*, p < 0.05; as determined by unpaired t-test.
(TIF)

**S12 Fig. Functional domains found in TEP20.** (**A**) Schematic representation of TEP20 functional domains predicted using the pfam webserver (http://pfam.xfam.org). (**B**) Localization of the thioester domain in TEP20 and its absence in AaMCR. Sequences were aligned with Clustal Omega (https://www.ebi.ac.uk/Tools/msa/clustalo/)).
(TIF)

# Acknowledgments

We thank Professor Eng Eong Ooi from Duke-NUS Medical School, for providing the DENV2 and ZIKV stock, and Professor Lisa Ng from Singapore Immunology Network (SIgN, A\*STAR, Singapore) for providing the CHIKV stock.

# Author Contributions

**Conceptualization:** R. Manjunatha Kini, Julien Francis Pompon.

**Formal analysis:** Avisha Chowdhury, Cassandra M. Modahl, Dorothée Missé, Thomas Vial, R. Manjunatha Kini, Julien Francis Pompon.

**Funding acquisition:** R. Manjunatha Kini.

**Investigation:** Avisha Chowdhury, Cassandra M. Modahl, Siok Thing Tan, Benjamin Wong Wei Xiang.

**Methodology:** Avisha Chowdhury, Cassandra M. Modahl, R. Manjunatha Kini, Julien Francis Pompon.

**Project administration:** R. Manjunatha Kini, Julien Francis Pompon.

**Supervision:** R. Manjunatha Kini, Julien Francis Pompon.

**Visualization:** Avisha Chowdhury, Cassandra M. Modahl.

**Writing – original draft:** Avisha Chowdhury, Cassandra M. Modahl, R. Manjunatha Kini, Julien Francis Pompon.

**Writing – review & editing:** Avisha Chowdhury, Cassandra M. Modahl, Siok Thing Tan, Benjamin Wong Wei Xiang, Dorothée Missé, Thomas Vial, R. Manjunatha Kini, Julien Francis Pompon.

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
