## [Decision Letter · Decision Letter 0]

10 Mar 2020

Dear Dr Pompon,

Thank you very much for submitting your manuscript "JNK pathway restricts DENV, ZIKV and CHIKV infection by activating complement and apoptosis in mosquito salivary glands" for consideration at PLOS Pathogens. As with all papers reviewed by the journal, your manuscript was reviewed by members of the editorial board and by several independent reviewers. In light of the reviews (below this email), we would like to invite the resubmission of a significantly-revised version that takes into account the reviewers' comments.

We cannot make any decision about publication until we have seen the revised manuscript and your response to the reviewers' comments. Your revised manuscript is also likely to be sent to reviewers for further evaluation.

Sincerely,

George Dimopoulos, PhD MBA

Guest Editor

PLOS Pathogens

Sara Cherry

Section Editor

PLOS Pathogens

Kasturi Haldar

Editor-in-Chief

PLOS Pathogens

orcid.org/0000-0001-5065-158X

Michael Malim

Editor-in-Chief

PLOS Pathogens

orcid.org/0000-0002-7699-2064

Reviewer's Responses to Questions

**Part I - Summary**

Reviewer #1: In this study, Chowdhury et al characterized the immune responses of Aedes aegypti salivary glands to dengue, Zika and chikungunya viruses, via both high-throughput transcriptomic assay and RNAi-based functional study. They identified that the JNK pathway, which can be induced by the three viruses infection, may regulate both complement response and apoptosis in salivary glands via TEP20 and Dronc, respectively, to play a common antiviral activity against DENV, ZIKV and CHIKV in Aedes aegypti salivary glands. Generally, this is an interesting study to offer an insight to the antiviral immunity of mosquito salivary glands, which may offer some potential targets to interrupt arboviral transmission. However, the authors should address the following scientific concerns before my positive recommendation.

Reviewer #2: In this manuscript the authors have carried out the experiments with great care and the results are supported by figures in great details. Using RNA-seq authors have done full-genome high-throughput transcriptomic analysis of the Aedes aegypti salivary glands upon Dengue, Zika, and Chikungunya viral infections. This comparative transcriptomic analysis has shown that the major components of Toll, IMD, and JNK pathways were regulated in the salivary glands upon viral infections. Authors have used RNAi-mediated gene silencing to further elucidate the possible mechanisms of JNK-pathway regulated viral defenses in the salivary glands. Finally by expression profiling of the key components of complement pathway and apoptosis in conjunction with the RNAi-mediated gene silencing of positive regulator of JNK pathway or co-silencing of these effectors together with the negative regulator Puckered, authors have concluded that the broad antiviral function of JNK pathway likely derives from the activation of complement and apoptosis pathways in the Aedes salivary glands. The manuscript is in general well prepared, however some of the major conclusions are over-stated. The results are generally clear and well-presented but in my opinion some key experiments are missing from the mechanistic studies of JNK-pathway directed anti-viral defenses that directly impact the novelty aspect of the manuscript. Overall, the transcriptomic analasis and RNAi assays are solid and robust, but at its current stage I feel it lacks the quality for PLoS Pathogens, unless these major and minor comments below are addressed:

**Part II – Major Issues: Key Experiments Required for Acceptance**

Reviewer #1: 1. The authors identified TEP20 as a resistant factor against the viral infections. Usually, TEPs play the anti-pathogen activity through a thioester domain. However, some study indicated a TEP-like factor without thioester domain, MCR, resists viral infection by regulating expression of AMPs. Does TEP20 have the thioester domain? A previous study showed that an antiviral AMP can be induced by DENV in the mosquito salivary glands (Luplertlop et al., 2011). Does TEP20 regulate the AMPs expression in SG?

2. There are many TEPs regulated by the viral infections in SG. For functional investigation, the authors silenced the TEP20 gene via dsRNA inoculation. Is it possible that the dsRNA TEP20 causes cross-silencing effect to other TEP genes? How about this silencing specificity?

3. A co-silencing assay suggests that both TEP20 and Dronc may function in the same antiviral pathway. I just wonder whether knockdown of TEP20 can really regulate the apoptosis in the mosquito salivary glands. The authors should provide more convincing evidence before this conclusion.

4. Line 145-191, there is too much pathway description in the Results. It is better to remove the most description words to the Discussion.

Reviewer #2: 1) Pg 5 Ln 117, the overlap of the DEGs from SG transcriptomes upon DENV, ZIKV, CHIKV is small, with only 19 DEGs are in common. Could this result be influenced by the infection intensities? Have the authors controlled the infection level, or checked the viral titer and infection prevalence by day 14 in the SGs? It’s interesting to see in Figure 1 the pathway analysis has suggested the genes been regulated by DENV from IMD and JNK pathways are very small numbers, however there are quite some genes were regulated upon CHIKV infection in the SGs. If JNK is a broad antiviral pathway, I would think similar genes or gene families should be regulated by all these three viruses. I believe it is very essential to measure the infection prevalence and intensities before sample preparations for RNA-seq assays. Authors have provided the information of the viral dosages in the blood meal, however there is no information of the viral titration in the SGs when they have collected samples for RNA-seq. In Figure 1, some of the DEGs were selected for further studies. What are the rationales authors have specifically selected these genes for further functional studies?

2) From RNA-seq based full-genome transcriptomic analysis authors have shown four TEPs are induced upon viral infections in the salivary glands. Furthermore, through qRT-PCR based expression analysis authors have shown that in the Kay gene silenced mosquitoes the expression of TEP20 and Dronc were significantly reduced. Therefore, authors have concluded that upon infection the JNK pathway induces the complement system through TEPs, and apoptosis through Dronc. This mechanistic study of the JNK-mediated antiviral defense in the salivary glands is overstated. Based on the results provided in this study, there is no direct result supporting this conclusion. First of all, authors haven’t provided experimental evidence to support that TEP20 is taking part in the complement pathway. Previous comprehensive study of insect TEPs and Aedes TEP1 protein (Xiao et al., 2014) has concluded that the complement-related protein control the flavivirus infection of Aedes mosquitoes through induction of antimicrobial peptides (AMPs). In that study authors have done phylogenetic analysis of A. aegypti macroglobulin complement-related factors (AsMCRs) and have shown that not all TEPs contain the TE module, and ones lack of this domain have a distinct mechanism of action. In that study, TEP20 was not characterized. In order to establish the link between JNK pathway, anti-viral defense, and complement pathway, authors have to do more detailed studies to show the evidence of TEP20 is indeed a complement-like factor in the Aedes mosquito anti-viral defense. Similar assays done by Xiao et al., 2014 could be applied to TEP20 in this study. Authors should assay the recognition of TEP20 to the tested viruses in vitro and in vivo like AaTEP1(AAEL006361). Another aspect is to study whether the recognition of TEP20 to either one of the tested viruses regulates the expression of AMPs. Without all these direct evidences, the conclusion of JNK pathway restricts viral infection by activating complement pathway in mosquito salivary glands is very vague.

3) Similarly, through RNA-seq and RNAi-mediated gene silencing and qRT-PCR assays authors have suggested that JNK pathway restricts DENV, ZIKV, and CHIKV infection by activating apoptosis. Again this conclusion is over-stated. Authors have shown through qRT-PCR that in the Kay gene silenced mosquitoes the expression of Dronc was significantly reduced, however authors jumped immediately to the conclusion that upon infection the JNK pathway induces the apoptosis through Dronc without further direct evidences. I would suggest authors to perform tunnel staining in conjunction with RNAi, qRT-PCR assays to show the direct evidence supporting this hypothesis.

**Part III – Minor Issues: Editorial and Data Presentation Modifications**

Reviewer #1: (No Response)

Reviewer #2: Pg3 Ln63, the citation of Kistler et al., 2015 is not an appropriate reference for refractory mosquitoes.

Figure 1, Functional annotation of DEGs, the functional group named “BF” seems inappropriate, could some of the genes within this group be classified in the other functional groups, such as “MET”, or “PROT”, or “DIG”, etc?

Ln 153, “suggesting activation of RNAi”, this expression is inappropriate, should be “RNAi pathway or RNAi machinery”.

Ln154, PGRP and GNBP should be spelled out here when the first time authors have mentioned these molecules

Ln204, the gene silencing efficacy ranged from 35-85% in SGs. The functional studies based on 35% efficacy are irrelevant to the ones with 85% gene silencing efficacy. Authors should either delete the result of the functional studies with this level of gene silencing, or increase the gene silencing efficacy through increasing the dosage of dsRNA.

Fig. 2, It is more appropriate to present the median level of viral titers than the mean ± 95% C.I., considering the distribution of viral loads is not a normal distribution. The Mann-Whitney test and Chi-squared test should be used for the statistical analysis on the infection intensities and infection rate respectively.

Ln 457, label DENV serotype 2 as DENV2 is better to differentiate from other serotypes.

Ln 470, mosquitoes were left to feed for 1.5 h, is that too long to feed viral infected blood meal to mosquitoes?

Ln 472, the viral titers in the blood meal is very different. Does this cause the different expression profiles of immune genes in the SGs through RNA-seq analysis? I would suggest authors to measure the viral loads in the SGs when they did the RNA-seq assays.

Ln 473, the titration of the viruses in the cell line cultures were measured with plaque assay using BHK-21 cells. However, for the functional studies the quantification of virus genome RNA copies was measured using RT-PCR. It is known viral loa by RT-PCR doesn’t correlate with plaque assay results, and the viral loads estimated with the qRT-PCR method were on average much higher than the PFU levels. Besides the qRT-PCR based assays cannot distinguish among viable virus and gRNA template. Therefore I would suggest authors to titrate the viral loads through plaque assays.

PLOS authors have the option to publish the peer review history of their article (what does this mean?). If published, this will include your full peer review and any attached files.

Reviewer #1: No

Reviewer #2: No
---

## [Decision Letter · Decision Letter 1]

17 Jun 2020

Dear Dr Pompon,

Thank you very much for submitting your manuscript "JNK pathway restricts DENV2, ZIKV and CHIKV infection by activating complement and apoptosis in mosquito salivary glands" for consideration at PLOS Pathogens. As with all papers reviewed by the journal, your manuscript was reviewed by members of the editorial board and by several independent reviewers. The reviewers appreciated the attention to an important topic. Based on the reviews, we are likely to accept this manuscript for publication, providing that you modify the manuscript according to the review recommendations.

Sincerely,

George Dimopoulos, PhD MBA

Guest Editor

PLOS Pathogens

Sara Cherry

Section Editor

PLOS Pathogens

Kasturi Haldar

Editor-in-Chief

PLOS Pathogens

orcid.org/0000-0001-5065-158X

Michael Malim

Editor-in-Chief

PLOS Pathogens

orcid.org/0000-0002-7699-2064

Reviewer Comments (if any, and for reference):

Reviewer's Responses to Questions

**Part I - Summary**

Reviewer #1: (No Response)

Reviewer #2: I appreciate that authors have addressed most of my major concerns and questions in the revised manuscript. No major comments at this time.

**Part II – Major Issues: Key Experiments Required for Acceptance**

Reviewer #1: (No Response)

Reviewer #2: (No Response)

**Part III – Minor Issues: Editorial and Data Presentation Modifications**

Reviewer #1: (No Response)

Reviewer #2: Ln144 and 145, spell out gene names DCR2 and Ago2 for the first time.

Pg9, Table 1. The color or the RNA-seq based expression induction may be coded with the fold-changes. For instance, the fold-change of “Induced” column could be indicated in the same Table cell.

Ln 307, Reference Sim et al 2012 should be numbered as other references. Similarly in Ln 312 give the reference number of Sim et al.

Ln 409, no need to add “the” to Professor

Figure legends, p-value<0.05 maybe simplified as p<0.05 and so for p<0.01, p<0.001.

Ln 911, S2 Fig. edgerR should be “edgeR”

PLOS authors have the option to publish the peer review history of their article (what does this mean?). If published, this will include your full peer review and any attached files.

Reviewer #1: No

Reviewer #2: No
---

## [Editor Report · Decision Letter 2]

26 Jun 2020

Dear Dr Pompon,

We are pleased to inform you that your manuscript 'JNK pathway restricts DENV2, ZIKV and CHIKV infection by activating complement and apoptosis in mosquito salivary glands' has been provisionally accepted for publication in PLOS Pathogens.

Best regards,

George Dimopoulos, PhD MBA

Guest Editor

PLOS Pathogens

Sara Cherry

Section Editor

PLOS Pathogens

Kasturi Haldar

Editor-in-Chief

PLOS Pathogens

orcid.org/0000-0001-5065-158X

Michael Malim

Editor-in-Chief

PLOS Pathogens

orcid.org/0000-0002-7699-2064
---

## [Editor Report · Acceptance letter]

31 Jul 2020

Dear Dr Pompon,

We are delighted to inform you that your manuscript, "JNK pathway restricts DENV2, ZIKV and CHIKV infection by activating complement and apoptosis in mosquito salivary glands," has been formally accepted for publication in PLOS Pathogens.

Best regards,

Kasturi Haldar

Editor-in-Chief

PLOS Pathogens

orcid.org/0000-0001-5065-158X

Michael Malim

Editor-in-Chief

PLOS Pathogens

orcid.org/0000-0002-7699-2064